# The Evidence of the Bystander Effect after Bleomycin Electrotransfer and Irreversible Electroporation

**DOI:** 10.3390/molecules26196001

**Published:** 2021-10-02

**Authors:** Paulius Ruzgys, Neringa Barauskaitė, Vitalij Novickij, Jurij Novickij, Saulius Šatkauskas

**Affiliations:** 1Biophysical Research Group, Vytautas Magnus University, Vileikos st. 844404, LT-44001 Kaunas, Lithuania; paulius.ruzgys@vdu.lt (P.R.); neringa.barauskaite@stud.vdu.lt (N.B.); 2Institute of High Magnetic Fields, Vilnius Gediminas Technical University, Naugarduko st. 4103227, LT-10224 Vilnius, Lithuania; vitalij.novickij@vgtu.lt (V.N.); jurij.novickij@vgtu.lt (J.N.)

**Keywords:** electroporation, bystander effect, electrochemotherapy, bleomycin, electroablation, irreversible electroporation, electrotransfer

## Abstract

One of current applications of electroporation is electrochemotherapy and electroablation for local cancer treatment. Both of these electroporation modalities share some similarities with radiation therapy, one of which could be the bystander effect. In this study, we aimed to investigate the role of the bystander effect following these electroporation-based treatments. During direct CHO-K1 cell treatment, cells were electroporated using one 100 µs duration square wave electric pulse at 1400 V/cm (for bleomycin electrotransfer) or 2800 V/cm (for irreversible electroporation). To evaluate the bystander effect, the medium was taken from directly treated cells after 24 h incubation and applied on unaffected cells. Six days after the treatment, cell viability and colony sizes were evaluated using the cell colony formation assay. The results showed that the bystander effect after bleomycin electrotransfer had a strong negative impact on cell viability and cell colony size, which decreased to 2.8% and 23.1%, respectively. On the contrary, irreversible electroporation induced a strong positive bystander effect on cell viability, which increased to 149.3%. In conclusion, the results presented may serve as a platform for further analysis of the bystander effect after electroporation-based therapies and may ultimately lead to refined application of these therapies in clinics.

## 1. Introduction

Enabled by more detailed characterization of new molecular tumor therapy features, facilitated and targeted drug delivery often is the desired option for tumor treatment [1]. Among many physical and chemical targeted drug delivery methods, the electroporation technique has proved to be one of the most efficient, leading to its adoption in clinical practice [2,3,4]. The phenomenon of electroporation occurs when cells or tissues are exposed to an external electric field, resulting in an increase in the transmembrane voltage across the plasma membrane. When the cellular transmembrane potential reaches the level of the electroporation threshold (in the range of 0.4 to 0.9 V), then pores of a hydrophilic nature start forming in the cell plasma membrane [5]. This enables the diffusion of various exogenous molecules into the electroporated cells [6]. The combination of membrane-impermeable anticancer drug administration and tumor electroporation led to the development of antitumor therapy termed as electrochemotherapy (ECT) [7,8]. Currently, ECT is a routine clinical treatment for cutaneous metastases of any histology and is listed as a primary skin cancer treatment for cutaneous metastases [9]. Bleomycin is the most frequently used anticancer drug for ECT, since electroporation of the selected areas of the tissue allows the use of this drug at a 1000 times lower concentration to trigger the desired killing effect [10]. The mechanism of action of bleomycin relies on the formation of hydroperoxyl radicals, which in turn create lesions of genomic DNA, resulting in apoptotic cell death [11].

It is also known that the electric current passing through the cells upon application of electric pulses generates reactive oxygen species (ROS) [12]. The association between partial membrane electroporation and the generation of ROS during the application of electric pulses has been previously reported [13,14].

Even though ECT is considered to be a local treatment, it is also known that ECT triggers systemic reactions associated with activation of the immune response [15]. One of the ways for activation of the immune response is related with damage-associated molecular patterns (DAMPs) [16]. The main DAMPs are release of ATP or high-mobility group box 1 (HMGB1) protein from the cells and externalization of membrane protein calreticulin (CRT) [17,18,19]. All these DAMPs have been reported to be involved after the ECT treatment [20].

As an alternative to ECT, efficient tumor treatment can be achieved by the use of irreversible electroporation, a local anticancer therapy without any anticancer drug used. In this case, cell death occurs because of dramatic disruption of cell homeostasis, leading to cell apoptosis or necrosis [21,22]. This can occur due to irreversible membrane disruption and ROS-associated lipid membrane oxidation [23]. Similar to ECT, IRE can induce immune system response by triggering the release of DAMPs and subsequent dendritic cell activation [21]. Interestingly, the rate and magnitude of the IRE-induced release of DAMPs, namely ATP and HMGB1, were reported to be much greater than those induced by other treatment modalities, such as ionizing radiation and chemotherapy [21]. Our previous study also demonstrated that cell death following cell electroporation is at least partially related to leakage of intracellular molecules [24].

In this context, both ECT and IRE are similar to ionizing radiation therapy [25]. The main mechanism of radiation therapy is the generation of reactive oxygen species (ROS) in the targeted tissue [26,27] and subsequent formation of single- or double-strand DNA breaks. Radiation-induced effects have also been shown to be related to DAMP generation, which in turn can significantly affect non-targeted cells [26,27]. The cells not directly targeted by ionizing radiation also suffer from chromosome aberrations, mutations, various epigenetic changes, DNA damage, formation of micronuclei, mitotic arrest, or apoptotic cell death [28,29,30,31,32]. Such effects on the untreated cells are known as bystander effect [29].

Since ECT and IRE share similar features of ROS generation and release of DAMPs with irradiation therapy, the bystander effect after ECT and IRE is also likely. In the present study, we aimed to investigate the role of the bystander effect following ECT and IRE.

## 2. Results

In most cases, 20 nM bleomycin concentration for electrotransfer into cells is a good compromise, since this concentration does not affect non-electroporated cells but induces a considerable cytotoxic effect on electroporated cells [33,34,35]. Therefore, we chose a bleomycin concentration of 20 nM. Firstly, we performed control experiments as shown in Figure 6. Cells were treated with bleomycin or electric pulse (direct treatment) and with a medium taken from untreated cells (control of indirect treatment) or cells treated with bleomycin or electric pulse (indirect treatment; Figure 1). Consistent with previous reports, cell treatment with electric pulse (1400 V/cm, 100 µs) or 20 nM of bleomycin had no significant effect on cell viability compared to the control. Conversely, when the medium was taken from the control cells or cells treated with electric pulse or bleomycin and transferred to untreated cells (indirect treatment), cell viability decreased to 72.35% ± 2.65% (*p* < 0.05), 83.25% ± 1.42% (*p* > 0.05) and 67.07% ± 3.23% (*p* < 0.05), respectively (Figure 1A). This result was unexpected; however, it can be explained by the fact that the medium was slightly depleted and ran out of some ingredients needed to maintain cell viability. The colony size showed no significant differences when comparing all experiment points with the control (Figure 1B).

Similar experiments were performed when treating cells with various bleomycin concentrations (in the range of 1–20 nM), followed by an application of a single electric pulse (1400 V/cm, 100 μs; Figure 2). The viability of treated cells decreased to 63.95% ± 4.63% with 1 nM bleomycin. Furthermore, cell treatment with 5 nM or a higher concentration of bleomycin resulted in cell viability decrease to about 2% (Figure 2A, direct treatment). Bleomycin electrotransfer also negatively affected cell colony size: at 1 nM bleomycin concentration, cell colony size decreased to 80.95% ± 2.16%; at 5 nM or a higher bleomycin concentration, cell colony size decreased to about 21% (Figure 2B, direct treatment).

The investigation of cell viability and colony size dependence on bleomycin concentration in the media taken from directly treated cells and applied on unaffected cells was performed in the same way. At all bleomycin concentrations, indirect treatment (1–20 nM) decreased cell viability in a concentration-dependent manner (Figure 2A, indirect treatment). At the highest bleomycin concentration of 20 nM, cell viability decreased to about 2%—the same as after the direct (bleomycin electrotransfer) treatment. It should be noted that cell incubation with the medium taken from directly affected cells with the highest bleomycin concentration (20 nM) and without electroporation did not affect cell viability compared to the cells grown in the affected medium (indirect effect) with no bleomycin. This indicates the insignificant effect of this concentration of bleomycin on cell viability without application of electric fields (see Figure 1).

The indirect cell treatment showed a different effect on colony size. Cell colony size decreased to some extent but did not exceed 80% at the bleomycin concentrations of 1, 5, and 10 nM. Further increase in bleomycin concentration resulted in a steep decline of cell colony size to about 30% (Figure 2B).

Strikingly, indirect treatment with a medium taken from cells treated using bleomycin electrotransfer had a dramatic effect on cell viability, especially at 20 nM. Therefore, we decided to investigate cell viability and cell colony size when treating cells with the affected medium, diluted at various proportions. For this, the medium was taken from cells treated with 20 nM bleomycin electrotransfer and diluted with growth medium to obtain the ratio of the affected medium ranging from 0 to 100% (Figure 3).

As expected, an increase in the dilution of the medium resulted in a rise in cell viability. A significant difference from the cell viability in the control group was observed when cells were grown in a medium containing 80% or 100% of the affected medium. At these dilutions, the viability of cells decreased to 25.02% ± 11.3% and 2.8% ± 3.37%, respectively (Figure 3A). Similarly, we also observed increases in cell colony size at these dilutions. A significant difference from control was observed when cells were grown in a medium containing 60%, 80%, or 100% of the affected medium. At these dilutions, cell colony sizes decreased to 41.78% ± 7.43%, 24.47% ± 3.8%, and 19.17% ± 6.9%, respectively (Figure 3B).

After observing the effect of indirect cell treatment with the medium taken from cells treated using bleomycin electrotransfer, we decided to check whether the medium taken from irreversibly electroporated cells has a similar effect. To achieve irreversible electroporation, the cells were electroporated with one pulse of 2800 V/cm strength and 100 µs duration (Figure 4).

As seen in Figure 4, a 2800 V/cm, 100 µs duration pulse decreased cell viability and cell colony size to 8.23% ± 8.99% and 78.15% ± 22.88%, respectively (shown as red lines). It is important to note that although cell viability changed dramatically, the changes in cell colony size were less pronounced.

In subsequent experiments, we decided to investigate cell viability and cell colony size after indirect treatment when cells were treated with a medium taken from irreversibly electroporated cells, at different dilution rates (Figure 4). Unexpectedly, the undiluted medium increased cell viability by 50% (*p* < 0.05). This increase was present for the ratios of the affected medium ≥40%. Only when the ratio of the affected medium was decreased below 20% did cell viability start to level to the control (Figure 4A). Cell colony size change had a different trend. Even though statistically significant (*p* < 0.05) differences were obtained with 80% and 100% of the affected medium, an insignificant decrease of the colony sizes, up to 15%, was present at other dilution ratios (Figure 4B).

## 3. Discussion

In this study, we analyzed the effect of indirect cell treatment when cells were grown in a medium taken from cells treated using either bleomycin electrotransfer or irreversible electroporation. Strikingly, this indirect cell treatment had a dramatic effect on cell clonogenic viability and cell colony size. Indeed, viability and colony size of the cells grown in the medium taken from the cells treated using bleomycin electrotransfer (at 20 nM bleomycin concentration) decreased to around 3% and 20%, respectively. By contrast, the viability of the cells that were grown in the medium taken from the cells after irreversible electroporation increased to 150% versus control.

The terms of direct and indirect cell treatment are in use in the field of radiation therapy, with targeted cells affected by direct irradiation and cells in close proximity to the irradiation zone undergoing indirect treatment. This led to the introduction of bystander effect, defined as biological effects: DNA damage; chromosomal instability; mutation; and occurrence of apoptosis in cells near the irradiated zone [36,37]. The bystander effect can alter the dynamic equilibrium between proliferation, apoptosis, quiescence, or differentiation. Similarly, the indirect effects after bleomycin electrotransfer as well as irreversible electroporation can also be associated with the bystander effect. It is important to note that bleomycin electrotransfer is closely associated with antitumor electrochemotherapy (ECT), an anticancer treatment modality that is currently implemented in clinics over the world as a local anticancer treatment therapy [38,39]. A well-known process of bleomycin-induced ROS generation in the presence of metal ions, like Cu^+^ or Fe^2+^, and oxygen, results in cellular DNA degradation [40,41]. It is also known that one bleomycin molecule can make multiple DNA lesions [42]. In our recent publication, we demonstrated that DNA degradation takes place after bleomycin electrotransfer at concentrations used in the present paper [43]. Since ECT, like radiation therapy, is a local treatment associated with local cell damage, ROS generation, DNA damage (in case of bleomycin electrotransfer), and release of DAMPs, the bystander effect following ECT seems to be very plausible (Figure 5).

Several mechanisms involving secreted soluble factors, oxidative metabolism, and gap–junction intercellular communication have been proposed to regulate the radiation-induced bystander effect [44]. These molecules (soluble factors) include nitric oxide (NO), transforming growth factor β1 (TGF-β1), COX-2, tumor necrosis factor α (TNF-α), and interleukins [31,45,46,47]. One can assume that the same molecules may be involved in the bleomycin electrotransfer-induced bystander effect, since the cellular responses are similar to those observed after ionizing radiation [48]. The investigation of the nature of these molecules was not the subject this study and will be analyzed in future papers.

The negative bystander effect can be in part explained as a result of the induction of mitotic arrest. Interestingly, the phenomenon of mitotic cell arrest in indirectly treated cells is also present after ionizing radiation [49], suggesting that the bystander effect after bleomycin electrotransfer and ionizing radiation may be governed by similar mechanisms. On the contrary, the impact of the bystander effect on cell viability and cell colony size is positive after the application of irreversible electroporation. The opposite direction of the bystander effect after irreversible electroporation shows that the mechanism is also different. Compared to the control, the bystander effect after irreversible electroporation resulted in the cell viability increasing to ~150%. Evidently, this can only be obtained by some factors in the media that promote both cell viability and cell proliferation. Under normal conditions, around 60% of plated cells are capable of forming colonies [50]. In our case, around 300–350 colonies in the control group form from 400 plated cells (plating efficiency around 75–87%), yet the bystander effect after irreversible electroporation resulted in around 150% of cell viability. Thus, the factors promoting cell viability should indeed be released in the medium from irreversibly electroporated cells. However, an additional hypothesis could be related to enhanced cell migration when particular cytokines are present in the extracellular medium [51,52]. Therefore, the factors released from irreversibly electroporated cells presumably play several roles, including promotion of cell viability and enhancement of cell migration. Similar results were obtained by our recent study, where we showed that some irreversibly electroporated cells can be rescued by supplementing the medium with compounds obtained from irreversibly electroporated cells [24]. We determined that the intracellular molecules that contribute to the increase in cell viability are larger than 30 kDa. A more detailed investigation of such molecules, as well as the factors that release in the medium after bleomycin electrotransfer, will be performed in subsequent studies.

The bystander effect after irreversible electroporation was analyzed in a recent study [53]. Contrary to our results, this study reported a slight negative bystander effect after IRE, showing that the viability of B16 cells, but not CMeC-1 cells, decreased by around 16%. Although the real nature of this discrepancy is not known, it can presumably be explained by the methodology of experiments. The medium from the affected cells was taken from one well (one experiment point) 24 h post electric field application and added to the wells with unaffected cells. Cell viability was measured after 24 h with Presto Blue reagent [53]. In our experiments, the affected medium was collected from 10 wells (10 experiment points) after 24 h of incubation, and the viability was evaluated by using the clonogenic assay 6 days after the treatment. Despite the unclear nature of this discrepancy, an important finding from the aforementioned study suggests that the signals accounting for the bystander effect could at least in part be mediated through extracellular microvesicles.

One of the critical issues on the local cancer treatment is that all targeted cancer cells must be affected significantly, making precise targeting a key factor [54]. However, this remains a challenge up to now [55]. Therefore, the bystander effect can propagate the treatment effect on the tumor cells that survived direct treatment and are close to directly affected cells. The results in this study give a broader view about the processes of electrochemotherapy, which might trigger a strong negative bystander effect similar to that shown in Figure 3 and Figure 4. On the other hand, antitumor treatment with irreversible electroporation can result in the induction of a positive bystander effect, which might eventually lead to the promotion of cell proliferation and/or cell migration (see Figure 4). Considering the potential importance of these results, the bystander effect after bleomycin electrotransfer and irreversible electroporation must be further explored in both in vitro and in vivo settings.

In conclusion, for the first time, we showed the presence of a negative bystander effect after bleomycin electrotransfer. We hypothesize that the negative bystander effect could be triggered by a similar mechanism as the one induced by ionizing irradiation. We also showed that irreversible electroporation can trigger a positive bystander effect. This effect is most likely related with the release of specific molecules from permeabilized cells; the composition of these molecules is presumably different from that of those occurring after bleomycin electrotransfer. The results of this study may serve as a starting point for further analysis of the bystander effect after electroporation-based therapies and may ultimately lead to the refined application of therapies in clinics.

## 4. Materials and Methods

### 4.1. Cell Culture

Chinese hamster ovary cells (CHO-K1) were used for the experiments. DMEM–high glucose (Sigma-Aldrich, St. Louis, MO, USA) medium supplemented with 1% penicillin-streptomycin (Sigma-Aldrich) and 10% of fetal bovine serum (FBS; Sigma-Aldrich) was used as a growth medium. The CHO-K1 cells were passaged every third day to ensure that cell confluence would not reach more than 80%. Cell passaging was performed using a trypsin-EDTA solution (Sigma-Aldrich) after pre-washing with phosphate buffer saline (PBS). After cell incubation in the trypsin-EDTA solution for 2 min at 37 °C, the cells were collected by centrifugation. The cells were then used for experiments or seeded in 96 mm^2^ tissue culture plates (TPP) with 10 mL growth medium and grown in a 5% CO_2_ humidified incubator at 37 °C.

### 4.2. Cell Electroporation

After trypsinization, the cells were suspended in a laboratory-made electroporation buffer (Na_2_HPO_4_ 5.59 mM, NaH_2_PO_4_ 3.00 mM, MgCl_2_ 1.73 mM, sucrose 242.2 mM) at a concentration of 2 × 10^6^ cells/mL. The conductivity of the electroporation media was 0.1 S/m with pH 7.3. Afterwards, 45 μL of prepared cell suspension was mixed with 5 μL anticancer drug bleomycin (10–200 nM) or 5 μL electroporation buffer for a control group. The resulting 50 μL of cell suspension containing 9 × 10^5^ of cells and 20 nM of bleomycin was then transferred in between laboratory-made stainless steel plate electrodes with a 2 mm gap and electroporated using one 100 µs duration square wave pulse of 1400 V/cm strength (for bleomycin electrotransfer) or 2800 V/cm strength (for irreversible electroporation) using a BTX T820 electroporator (Harvard Apparatus).

### 4.3. Cell Viability Evaluation

#### 4.3.1. Viability Evaluation of Directly Affected Cells after the Bleomycin Electrotransfer or Irreversible Electroporation

After bleomycin electrotransfer or irreversible electroporation, the affected cell suspension was placed in a 1.5 mL Eppendorf tube and incubated for 10 min. Then, the cells were diluted with the DMEM growth medium and 400 cells were plated in a 40 mm Petri dish (TPP) with 2 mL DMEM growth medium and transferred into the incubator (5% CO_2_ at 37 °C) for 6 days. Afterwards, the colonies formed were stained with crystal violet solution (40% ethanol, 20% distilled water, 40% crystal violet dye (Sigma-Aldrich)). The number and size of the colonies was evaluated using digital images and ImageJ software (National Institute of Health, Bethesda, MD, USA) following recommendations provided by the creators of the software [56,57] and compared to the number and size of the colonies in the untreated control.

#### 4.3.2. Viability Evaluation of Indirectly Affected Cells after Bleomycin Electrotransfer or Irreversible Electroporation

After bleomycin electrotransfer or irreversible electroporation, cell suspension was placed in a well of a 24-well plate (TPP) and supplemented with 0.2 mL of DMEM growth medium after 10 min incubation. The same experiment was repeated to fill 10 wells of the 24-well plate. Afterwards, the 24-well plate was transferred into the incubator (5% CO_2_ at 37 °C) for 24 h. Then, the growth medium was removed from the wells and centrifuged twice to remove possible detached cells. The resulting 2 mL supernatant of the growth medium (affected medium) was transferred to previously (24 h before) plated untreated 400 cells in a 40 mm Petri dish (TPP). If needed, the supernatant was diluted with a freshly prepared growth medium and transferred to previously plated untreated 400 cells. The Petri dishes were then transferred into an incubator (5% CO_2_ at 37 °C) for 5 days. Afterwards, the colonies formed were evaluated as described above. For simplicity, the experimental protocol for the evaluation of cell viability of directly and indirectly affected cells after bleomycin electrotransfer is shown in Figure 6.

### 4.4. Statistical Analysis

Statistical analysis was done with MS Excel and Prism 9 software. The data in the figures are represented as mean ± standard error of the mean (SEM). Statistically significant differences between experimental groups were evaluated using one-way analysis variance (one-way ANOVA) followed by the Bonferroni test. The significance was marked as *, **, or *** when the *p*-values were less than 0.05, 0.01, or 0.001, respectively.

## Figures and Tables

**Figure 1 molecules-26-06001-f001:**
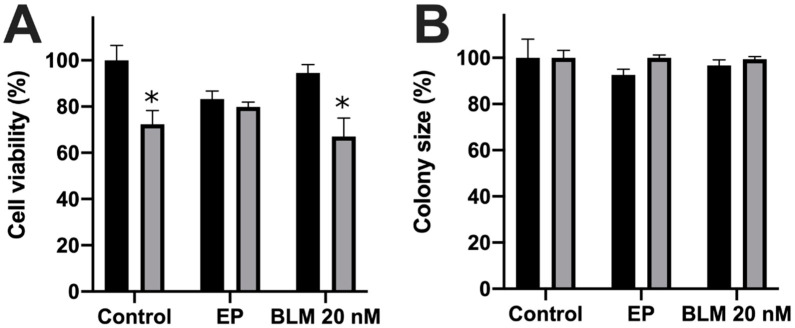
Cell viability (**A**) and cell colony size (**B**) changes after cell treatment with electric pulse (1400 V/cm, 100 µs) or bleomycin (20 nM) (direct treatment) or after cell incubation in the medium taken from control cells or directly treated cells (indirect treatment). The black and gray bars represent directly and indirectly treated cells, respectively. The * symbol represents statistical significance *p* < 0.05 compared to the control.

**Figure 2 molecules-26-06001-f002:**
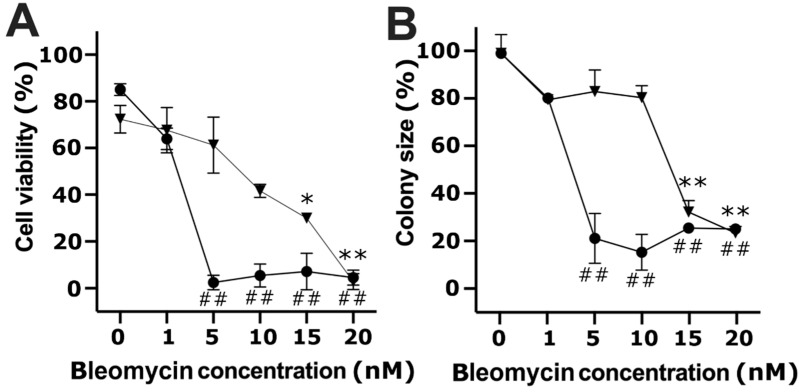
Cell viability (**A**) and cell colony size (**B**) changes after bleomycin electrotransfer (direct treatment) and after cell incubation in the medium taken from directly treated cells (indirect treatment) dependance on bleomycin concentration. For cell electroporation, a single 1400 V/cm strength and 100 μs duration electric pulse was used. The circles and triangles represent directly and indirectly treated cells, respectively. The * and ** or ## symbols represent statistical significance of *p* < 0.05 and *p* < 0.01, respectively, compared to the control.

**Figure 3 molecules-26-06001-f003:**
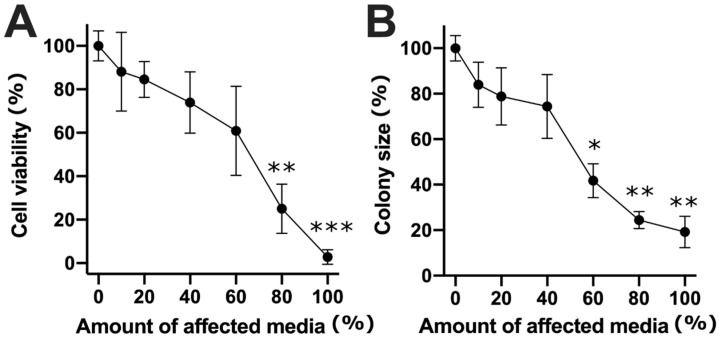
Cell viability (**A**) and cell colony size (**B**) changes after cell growth in a medium containing different ratios of the affected medium. The affected medium was taken from cells that were electroporated (one pulse 1400 V/cm, 100 μs) in the presence of 20 nM bleomycin. The *, **, and *** symbols represent statistical significance of *p* < 0.05, *p* < 0.01, and *p* < 0.001, respectively, compared to the control.

**Figure 4 molecules-26-06001-f004:**
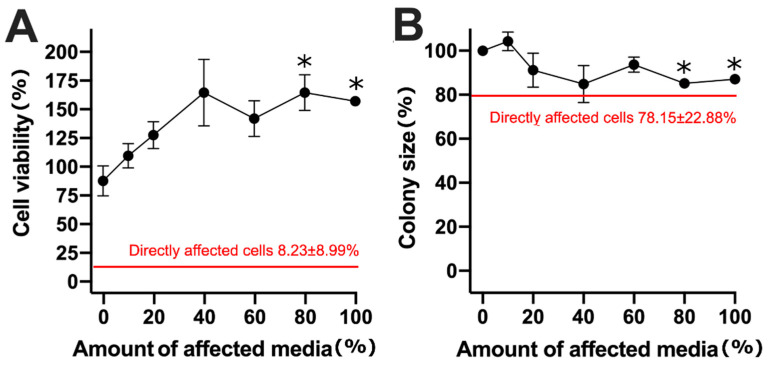
Cell viability (**A**) and cell colony size (**B**) changes after cell growth in a medium containing different ratios of the affected medium. The affected medium was taken from cells that were irreversibly electroporated by using one 2800 V/cm, 100 µs pulse. The red lines and black curve represent directly and indirectly affected cells, respectively. The * symbol represents statistical significance of *p* < 0.05 compared to the control.

**Figure 5 molecules-26-06001-f005:**
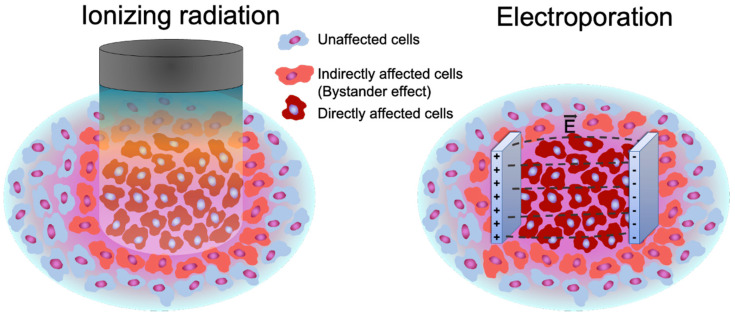
Visual representation of the bystander effect after ionizing radiation and electroporation trigger bleomycin electrotransfer. Three types of cells can be discriminated: directly affected cells, indirectly affected cells (bystander effect), and unaffected cells.

**Figure 6 molecules-26-06001-f006:**
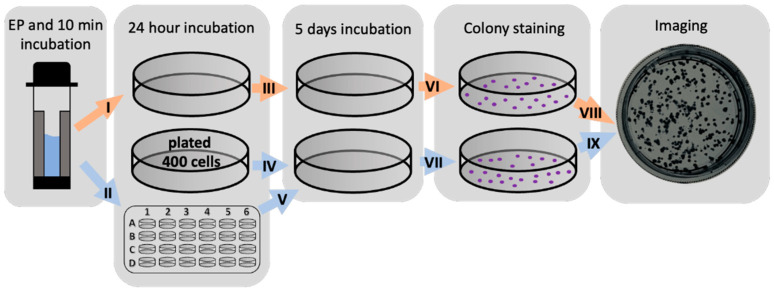
Visual representation of cell viability evaluation after direct (orange arrows) and indirect (blue arrows) treatments.

## Data Availability

The data presented in the study are available on the request from the corresponding author.

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
