# Peer review of "The Evidence of the Bystander Effect after Bleomycin Electrotransfer and Irreversible Electroporation"

_molecules, 2021, doi:10.3390/molecules26196001_

Round 1
Reviewer 1 Report
Please see the attached review report.

Author Response
Reviewer #1
Review comments on molecules-1352959 (The proof of bystander effect bleomycin electrotransfer and irreversible electroporation) The authors reported that the so-called bystander effect of medium culturing cells that treated with bleomycin electrotransfer or irreversible electroporation to CHO-K1 cells. This reviewer thinks that this version of manuscript cannot be accepted for publication. But a second consideration would be possible if the following issues can be reasonably addressed.
- The title is too general and misleading with grammar errors.
In the uploaded manuscript the title was: “The proof of bystander effect after bleomycin electrotransfer and irreversible electroporation”. If the title would be as the reviewer show in the comments, we completely agree that it would be “misleading with grammatical errors”.
Nevertheless, upon the reviewer’s notice, we have slightly modified the title. It now reads: The evidence of the bystander effect after bleomycin electrotransfer and irreversible electroporation.
- The manuscript should be proofread thoroughly for grammar errors, words missing, typos, and reference citation errors, such as (not limited to)the caption of Figure 1 seems not correct, line 94, ‘and’ is missing, Ref. [55]is not about the bystander effect after IRE…
Thank you for the comment. We have corrected the sentence in the line 94 and we have corrected the citation error. The ref [55] should have been ref [53]. We have carefully proofread the manuscript to clear out any additional typo errors.
- According to the discussion in the section of introduction, the authors believe that the ECT and IRE share the similar patterns of ROS generation and release of DAMPS with irradiation therapy. But the results in the cell viability and colony size of CHO-K1 cell show the totally different trends with the amounts of affected media treated by ECT and IRE. Explanation for this phenomenon provided in this manuscript is not sufficient and further clarification is still required.
Indeed, the trends of cell colony formation are different. This result was unexpected as well as the presence of the Bystander effect. Therefore, we believe that the results presented in our study are of a great interest. Nevertheless, at the current stage, the main purpose of the manuscript was to disclose the Bystander effect and show the differences of the Bystander effect after bleomycin electrotransfer and IRE.
Thanks to this feedback from the reviewer, we have updated the introduction to provide this message in more clear fashion.
- Any fresh medium was supplemented to cells during the 5-day incubation? How can we determine the change in the viability or colony size of CHO-K1 cell is due to the bystander effect instead of the depletion of nutrition in the culture medium?
We thank the reviewer for this important question. The medium was not supplemented with fresh medium during the 5-day incubation. To test that the observed effect is not caused because of depletion of nutrients in the culture medium, we performed control experiments when cells were treated with medium taken from control (untreated) cells. Although it had some negative effect (possibly due to depletion of nutrients in the culture) (see Fig. 1), significantly greater decline of cell viability and colony size after cell incubation in medium taken from cells treated with bleomycin electrotransfer (see Figs. 2 and 3) suggests the Bystander effect.
- What is the response of viability or colony size of CHO-K1 cell to the medium treated with IRE?
The answer to the reviewer’s question can be found in the results presented in Fig. 4. As one can see, the medium taken from irreversibly electroporated cells increased cell viability (the number of colonies) but slightly decreased the colony size.
- Can we determine the releases of cells treated with ECT and IRE? The result may be able to explain the difference in viability and colony size between ECT and IRE.
Thank you for the question. We agree with the reviewer that the most plausible explanation of the Bystander effect differences after ECT and IRE are related with the composition of molecules released into the medium after the treatments. There are almost no publications on ECT and IRE that analyze these molecules. Currently we can only speculate of the nature of these molecules. We agree with the reviewer that this is interesting and important question, and definitively deserves a separate study.

Reviewer 2 Report
I suggest a rejection of this manuscript.
(1) Authors should complete their control experiment first.
(2) The co-incubation system such as a transwell plate should be used for investigating cell-cell communication.
(3) Any molecular studies including cytokines should be supported for conclusion of bystander effect.
Author Response
I suggest a rejection of this manuscript.
(1) Authors should complete their control experiment first.
Thank you for this important notice. We believe that we performed all controls needed to disclose the Bystander effect. Unfortunately, the reviewer does not state the controls that should be additionally shown in the control experiments. Since the reviewer did not specify which controls would be needed, we can only speculate that electroporation-only controls after direct and indirect cell treatments were suggested. We have updated the Fig. 1 A and B with these additional data.
(2) The co-incubation system such as a transwell plate should be used for investigating cell-cell communication.
Thank you for this suggestion. Indeed, the transwell plate inserts could provide additional supporting information towards the described Bystander effect after ECT and IRE. We will use the reviewer’s advice in future research. However, we believe that the proof of the Bystander effect after ECT and IRE is the main aim at the current stage of the research. Therefore, we believe that standard protocols used to analyze the Bystander effect after radiotherapy, such as clonogenic assay, are fully sufficient.
(3) Any molecular studies including cytokines should be supported for conclusion of bystander effect.
We agree with the reviewer that cytokine studies would help to analyze the mechanisms of the Bystander effect. These types of experiments are planned for the future studies. Nevertheless, at the current stage, the main objective of this study was to demonstrate the cell death triggered by the Bystander effect, rather than investigating its mechanisms.

Reviewer 3 Report
Ruzgys et.al., in their manuscript titled, "The proof of Bystander effect after bleomycin electrotransfer and irreversible electroporation" evaluate bystander effect following electrochemotherapy and electroablation treatments. In this study they utilized the CHO cell model to measure viabilities and colony sizes by using cell culture medium from directly treated cells and then applying it on unaffected cells. The presented results suggest that bleomycin electrotransfer had a strong negative impact both on cell viability and cell colony size. On the other hand, electroporation induced a strong positive bystander effect on cell viability.
While the authors have made a sincere effort in presenting their results on an important topic, pertinent questions if addressed can strengthen their argument:
- Why was the CHO cell lines used?
- Can the authors recapitulate the CHO cell observations in an independent cell line? preferably a human cell line?
- It is of interest to see how ROS differs in the two treatment plans? there are several kits available to measure these.
- It is also of interest to know whether another double strand break inducing agent, such as etoposide might fare within the same treatment module.
- the conclusions to this paper can also be strengthened by quantifying bystander effect by measuring comet tails, or H2AX phosphorylation.
Author Response
Ruzgys et.al., in their manuscript titled, "The proof of Bystander effect after bleomycin electrotransfer and irreversible electroporation" evaluate bystander effect following electrochemotherapy and electroablation treatments. In this study they utilized the CHO cell model to measure viabilities and colony sizes by using cell culture medium from directly treated cells and then applying it on unaffected cells. The presented results suggest that bleomycin electrotransfer had a strong negative impact both on cell viability and cell colony size. On the other hand, electroporation induced a strong positive bystander effect on cell viability.
While the authors have made a sincere effort in presenting their results on an important topic, pertinent questions if addressed can strengthen their argument:
1. Why was the CHO cell lines used?
Thank you for the question. We have chosen CHO-K1 cells because it is the most frequently used cell line in the field of electroporation.
2. Can the authors recapitulate the CHO cell observations in an independent cell line? preferably a human cell line?
We agree that different cell lines would enhance the observed effect. This is planned for the future studies that are already in progress. The results are supported by our unpublished results on murine breast cancer cell line (4T1) showing similar results after IRE. Therefore, the Bystander effect is not limited to CHO cells only.
3. It is of interest to see how ROS differs in the two treatment plans? there are several kits available to measure these.
We agree with the reviewer that ROS can be measured with several kits. However, it is partly out of scope, since we believe that the Bystander effect is induced by the molecules released from the affected cells rather than ROS in the medium or in directly affected cells. The ROS generation in directly affected cells is already known and published with irreversible electroporation (Szlasa et. al. 2020) and bleomycin internalized in the cells (Allawzi 2019, Wallach-Dayan 2006). At this stage, the primary goal of our research is to prove the presence of the Bystander effect after ECT and IRE. The analysis of the Bystander effect mechanisms is planned in separate future studies.
4. It is also of interest to know whether another double strand break inducing agent, such as etoposide might fare within the same treatment module.
We thank the reviewer for this idea. That indeed would be a great positive control. We will consider implementing this positive control in future experiments when analyzing the mechanisms of the Bystander effect.
5. the conclusions to this paper can also be strengthened by quantifying bystander effect by measuring comet tails, or H2AX phosphorylation.
To the best of our knowledge, comet assay or H2AX phosphorylation is evaluating DNA strand breaks or their repair. It would be good for the evaluation of DNA strand breaks in directly affected cells. We have already published some data in this area (see Chopra et. al. 2019). However, it is not known whether the Bystander effect induces DNA strand breaks on indirectly affected cells. As mentioned above, systemic analysis of the mechanisms of the Bystander effect after ECT and IRE are foreseen as separate studies.

Round 2
Reviewer 1 Report
I appreciate the responses from the authors to my questions in the first round of reviewing. I recommend that the revised version of manuscript be accepted for publication in Molecules. One extra suggestion (might be helpful for the future submissions for the authors): revisions in the new submission might be highlighted that would make the reviewing more easy and friendly to peer reviewers.
Reviewer 3 Report
Thank you for your responses.